# Field Investigation Evaluating the Efficacy of Porcine Reproductive and Respiratory Syndrome Virus Type 2 (PRRSV-2) Modified Live Vaccines in Nursery Pigs Exposed to Multiple Heterologous PRRSV Strains

**DOI:** 10.3390/ani15030428

**Published:** 2025-02-04

**Authors:** Sunit Mebumroong, Hongyao Lin, Patumporn Jermsutjarit, Angkana Tantituvanont, Dachrit Nilubol

**Affiliations:** 1Swine Viral Evolution and Vaccine Development Research Unit, Department of Veterinary Microbiology, Faculty of Veterinary Science, Chulalongkorn University, Bangkok 10330, Thailand; sunit.dvm@gmail.com (S.M.); jermsutjarit@gmail.com (P.J.); 2MSD Animal Health Innovation Pte Ltd., Perahu Road, Singapore 718847, Singapore; hongyao.lin@msd.com; 3Department of Pharmaceutics and Industrial Pharmacy, Faculty of Pharmaceutical Sciences, Chulalongkorn University, Bangkok 10330, Thailand; tuvanont@gmail.com

**Keywords:** nursery pigs, PRRSV, modified live vaccines, heterologous PRRSV, protective efficacy

## Abstract

The objective was to understand the improvement in clinical parameters in piglets where different modified live vaccines (MLVs) against porcine reproductive and respiratory syndrome (PRRS) were administered. We conducted the study in the field with a large group of piglets and three different MLVs, assessing both safety and efficacy parameters. We found that different MLVs can be used at different stages of production if needed without adverse impacts on production and that there are differences in safety and efficacy between MLVs.

## 1. Introduction

Porcine reproductive and respiratory syndrome (PRRS) is a pig disease that is characterized by reproductive and respiratory disorders in breeding sows and growing pigs. Since its first emergence in the late 1980s, PRRS has consistently caused economic devastation in the swine industry worldwide. PRRS virus (PRRSV), an enveloped, positive-sense single-stranded RNA virus belonging to the *Arteriviridae* family, order *Nidovirales*, is the causative agent of this disease. Attempts to mitigate the economic losses encountered in swine farms by PRRSV infection have mainly centered on the management of replacement gilts, with strategies including acclimatization prior to introduction and vaccination with modified live vaccines (MLV). Vaccination with killed vaccines induces a poor immune response, especially in naive unexposed animals [1]. MLVs against PRRS used in control programs include vaccination in replacement gilts prior to introduction, mass vaccination of the entire breeding herd every 3–4 months, and piglet vaccination at 2 weeks of age. Despite implementing such rigorous vaccination programs, many swine herds using this program continue to experience PRRS-related production losses, suggesting that further optimization of vaccination protocols is needed.

The PRRSV genome is approximately 15 kb in length and consists of 11 open reading frames (ORFs), including ORFs 1–7 [2]. ORF1, which represents approximately 80% of the whole genome, encodes replicative enzymes and is further divided into ORFs 1a and 1b. ORFs 2–7 encode structural proteins including glycoproteins (GP) 2–5, and M and N proteins. ORF5 contains decoy and primary neutralizing epitopes [3] and plays important roles in genetic variation and protection [4,5]. The ORF5 sequence of PRRSV is widely used to study phylogeny and genetic variation due to its highly variable nature. Based on ORF5 classification, PRRSV is divided into two species: *Betaarterivirus suid 1* or PRRSV-1 and *Betaarterivirus suid 2* or PRRSV-2 [6]. Due to their genetic divergence, PRRSV-1 and PRRSV-2 are further categorized into four subtypes and nine lineages, respectively [7,8].

A key component of vaccination protocols is the selection of the MLV vaccine. Several authors have advocated for MLV selection criteria based on “as close as possible” genetic or antigenic matching between field strains and the MLV strain [9,10]. This is commonly performed by determining genetic similarity based on ORF5 sequences. ORF5 encodes for a highly variable envelope protein, GP5, which has been considered a major target protein for vaccine design as it is involved in the production of neutralizing antibodies, followed by protection against PRRSV. In practical terms, this is increasingly difficult to perform in the field for several reasons.

Firstly, the coexistence of PRRSV-1 and PRRSV-2 infection has been increasingly reported in Asia [11,12,13]. The presence of co-infection of PRRSV-1 and PRRSV-2 in farms complicates the selection of MLV vaccines, as the ORF5 sequences from these two genomes are highly dissimilar. Secondly, multiple PRRS MLVs are commercially available in Asia and based on strains such as VR2332, NEB-1, HuN4-F112, JXA1-P80, CH-1R or R98. The strains VR2332, CH-1R, and R98 are of lineage 5 [14], while NEB-1 is of lineage 7 [15] and the others belong to lineage 8 [16]. The high mutation rate of PRRS means that new isolates of PRRS are constantly emerging in the field through mutation or recombination. PRRSV has one of the highest mutation rates of RNA viruses. Different studies have suggested a mutation rate between 4.71 × 10^2^ and 9.8 × 10^2^/synonymous sites/year [17,18,19]. Recombination is also frequently seen in the field with PRRS isolates, either between field strains or between field and vaccine strains, often resulting in new PRRS genotypes emerging, sometimes with increased virulence [20,21]. Regulatory approval of new PRRS MLV vaccines can take years before commercial licenses are granted [22]. Consequently, this means that producers are forced to use existing commercially available vaccines to provide cross protection against emerging PRRS field strains and there is a need to evaluate existing vaccines on their ability to provide cross protection against contemporary circulating PRRS field strains.

Additionally, the evidence for using ORF5 matching strains is controversial. Previous studies have found that vaccination of swine with MLVs was able to induce good protection in pigs exposed to genetically diverse PRRSV strains post-vaccination [23,24]. Apart from genetic similarity, farms are also advised to select MLVs based on safety profile. New PRRSV strains can emerge on farms either due to biosecurity breaks, low fidelity during RNA virus replication, recombination events or random mutations [25]. By selecting a PRRSV vaccine that exhibits decreased shedding post-vaccination, the risk of recombination in the field with MLV and field isolates is reduced [26,27,28]. Producers have also been advised to ideally use only one MLV isolate for use within the entire production flow to prevent recombination events [29]; i.e., both sows and piglets should be vaccinated with the same MLV. This is again difficult to implement in practice as some farms may sell piglets to other farms or have piglets raised on separate sites, making it impossible to enforce the use of only one MLV across the entire lifecycle of production.

Therefore, the objectives of the present study were to evaluate the protective efficacy of three MLVs, based on PRRSV-2, in nursery pigs in a worst case scenario where the MLV does not match the genetic profile of the field isolate, different MLVs are used for sows and piglets, and piglets are naturally exposed to genetically distinct heterologous PRRSV isolates.

## 2. Materials and Methods

### 2.1. Ethical Statement for Experimental Procedures

All animal procedures were conducted in accordance with the Guide for the Care and Use of Laboratory Animals of the National Research Council of Thailand according to protocols reviewed and approved by the Chulalongkorn University Animal Care and Use Committee (protocol number 2031015).

### 2.2. Herd Information

This study was carried out on a commercial swine herd with an inventory of 10,000 sows in Thailand. The herd was selected with permission from the herd owner. The studied herd was a one-site farrow-to-finish production line with an internal replacement system. There were designated buildings for breeding, gestation, and farrowing activities. The farrowing facility operated all-in/all-out by week and allowed a week of downtime. All sows were artificially inseminated using PRRSV-negative semen from PRRSV-free boars. Semen was tested by PCR prior to insemination. Pigs were weaned at approximately 21 days of age and moved to nursery facilities 500 m away from the breeding facilities. Each nursery building operated all-in/all-out by week, and pigs from one farrowing house were weaned to one nursery house. Nursery pigs were housed for 8 weeks before moving to finishing facilities. Replacement gilts were produced internally and housed separately from nursery and finishing pigs. Internally produced gilts were moved to a gilt developing unit at 18 weeks of age and introduced to the breeding herd at 34 weeks of age. The acclimatization protocol in the gilt-developing unit included commingling for 4 weeks with culled sows, starting at 20 weeks of age, with a 10-week cool-down period. Gilts were serologically monitored for antibody responses by ELISA, and for viremia by PCR, with negative results required prior to introduction into the breeding herd.

The PRRSV status of the study herd was positive–stable–active [30]. The designated status refers to a serologically positive breeding herd and evidence of PRRSV circulation during the nursery period. In this study herd, seroconversion as measured by IDEXX ELISA was present in nursery pigs at 6 weeks of age. Both PRRSV-1 and PRRSV-2 were detected by PCR. The PRRS control program includes vaccination with PRRSV-2 MLV in which only one PRRSV-2 MLV (Ingelvac^®^ PRRS MLV, Boehringer Ingelheim, Ingelheim am Rhein, Germany) has been used in the herd for more than 5 years. The vaccination program includes quarterly whole-breeding-herd vaccination. Replacement gilts were vaccinated with two doses of the PRRSV-2 MLV (Ingelvac^®^ PRRS MLV, Boehringer Ingelheim, Ingelheim am Rhein, Germany) at 18 and 22 weeks of age. Piglets were intramuscularly vaccinated once at 2 weeks of age. In addition, vaccination against *M. hyopneumoniae* and PCV2 was routinely performed in all piglets at 3 weeks of age. Pigs in all treatment groups had ad libitum access to a high-quality, three-phase nursery feeding program. Each phase of the nursery feed included a combination of tiamulin and amoxicillin at concentrations of 150 and 400 ppm, respectively, to control bacterial infections. During the entire study, no changes in other management strategies potentially influencing the introduction of PRRSV into the herd were observed.

### 2.3. Experimental Design

Seventy-six thousand and seventy-five (*n* = 76,075), 2-week-old pigs from a sow herd that had been vaccinated quarterly with PRRSV-2 MLV (Ingelvac^®^ PRRS MLV (Boehringer Ingelheim, Ingelheim am Rhein, Germany) for more than 5 years were allocated into 4 groups, including US1-MLV (*n* = 51,535), US2-MLV (*n* = 1200), US3-MLV (*n* = 22,228), and NonVac (*n* = 1112) (Table 1). US1-MLV, US2-MLV, and US3-MLV were selected as these vaccines were commercially available to the company operating the farms and were in inventory at the time of the study and hence available to be used by the farms.

At 0 days post-vaccination (DPV), approximately 2 weeks of age, pigs in the US1, US2, and US3-MLV groups were intramuscularly vaccinated with different PRRSV-2 MLVs according to manufacturer’s instructions, including a 2 mL dose of Ingelvac^®^ PRRS MLV (Boehringer Ingelheim, Ingelheim am Rhein, Germany), a 2 mL dose of HuN4-F112 (Harbin Veterinary Research Institute, CAAS, Harbin, China), a PRRSV-2 lineage 8.7-based MLV, and an 1 mL dose of Prime Pac^®^ PRRS (MSD Animal Health, Rahway, NJ, USA), a PRRSV-2 lineage 7-based MLV, respectively. The NonVac group was left unvaccinated. Pigs in US1-MLV, US2-MLV, US3-MLV, and NonVac groups were weaned at 26 days of age and separated into 56, 2, 32, and 2 different nursery grow-outs, respectively. Two grow-outs of each group were randomly selected. Five pigs in each selected grow-out were randomly selected, and individually ear-tagged. The assignment of pigs into the nursery grow-outs was conducted by the farm management and the study monitors were not able to control this. For US2-MLV and NonVac groups, there were only 2 grow-outs available to use, so all were used for blood collection. For the US1-MLV and US3-MLV groups, 2 grow-outs were selected randomly for blood collection. Each grow-out was plotted onto a spreadsheet (Microsoft Excel^®^, Redmond, WA, USA) and a 20-sided dice was rolled. Only if the dice displayed 1 would the site be selected. If no sites managed to roll a 1, unselected sites would be assigned again until 2 sites were found. For the selection of pigs within the selected sites, each site would be filled with approximately 600 pigs. On arrival, pigs were divided into 30 pens of 20 pigs as they came off the transport. Study monitors would roll the 20-sided dice for every pen until a 1 was found. If a 1 was found, a pig would be selected within that pen to be ear-tagged for blood collection until 5 pigs were identified.

Blood samples were collected at 0, 14, 28, 42, and 56 DPV. Sera were separated and assayed for the presence of PRRSV and PRRSV-specific antibodies using PCR, ELISA, and serum neutralization (SN) assay, respectively. Positive PCR samples were subjected to PRRSV sequence. Peripheral blood mononuclear cells (PBMC) were isolated and analyzed for interleukin 10 (IL-10) using commercial enzyme-linked immunosorbent assay (ELISA) at 14 DPV, and interferon γ (INF-γ) using enzyme-linked immunosorbent spot (ELISPOT) at 35 DPV, respectively. A new sterile 1-inch, 18-gauge needle and syringe were used to collect a blood sample from each animal. Mortality and feed intake were recorded daily. Survival and average daily gain (ADG) rates were collected from company data records across all animals and grow-outs enrolled in the study. The person responsible for assessing the mortality, survival rate, feed intake, and ADG was the farm’s supervising veterinarian.

### 2.4. Vaccines and Viruses

Homologous and heterologous viruses were used to perform an SN assay. Homologous virus refers to a vaccine isolate. To isolate a homologous virus, each vaccine was reconstituted in DMEM. The virus was propagated in MARC-145 cells using a previously described method (Madapong et al. 2017) [31]. Virus was harvested by a cycle of freezing and thawing. The supernatant containing the virus was stored at −80 °C before subsequent use. Heterologous viruses refer to the field isolates of PRRSV-2 collected from the study herd from December 2017 to April 2020. Serum samples were collected every 4 months: December 2017, April 2019, August 2019, December 2019, and April 2020. At each sampling time, 5 blood serum samples were collected from nursery pig population. The sera were separated and assayed for the presence of viruses by polymerase chain reaction (PCR). Sequence reactions were performed at Biobasic Inc. (Markham, ON, Canada) using an ABI Prism 3730XL DNA sequencer. Pairwise sequence identity percentages were further assessed. A phylogenetic tree was constructed from aligned nucleotide sequences based on ORF5 genes of PRRSV-2 isolates. The heterologous viruses, THA_SP/RB_2783766/S3/17-7 and THA_SP/RB_S1/P1/0120-18 isolates, were isolated from nursery pigs in the farm displaying clinical signs associated with PRRS. The nucleotide and amino acid similarities (based on the ORF5 gene) between these Thai PRRSV-2 field isolates and PRRSV MLVs are summarized in Table 2.

### 2.5. PRRSV Detection

PRRSV in serum samples was detected using RT-PCR as previously described [32]. In brief, PRRSV RNA was extracted from serum samples using the NucleoSpin^®^ RNA virus kit (Macherey-Nagel Inc., Duren, Germany) in accordance with the manufacturer’s instructions. cDNA was synthesized from the extracted RNA using M-MuLV reverse transcriptase (New England BioLabs Inc., Hitchin, Hertfordshire, UK). PCR amplification was performed on cDNA using a commercial kit (GoTaq^®^ Green Master Mix, Promega, Woods Hollow Road, Madison, WI, USA), and to amplify ORF 5 of US progeny viruses, the following primers were utilized: ORF 5 USF (5′-CCT GAG ACC ATG AGG TGG G-3′) and ORF 5 USR (5′-TTT AGG GCA TAT ATC ATC ACT GG-3′). After the initial incubation at 95 °C for 2 min, the reactions were subjected to 35 cycles of PCR as follows: 95 °C for 30 s, 54 °C for 30 s, and 72 °C for 45 s, followed by a terminal, 5 min extension at 72 °C. Amplified PCR products were purified using a PCR purification kit (Macherey-Nagel, Valencienner Str. 11, Düren, Germany). Sequencing reactions were performed at Biobasic Inc. (Markham, ON, Canada) using an ABI Prism 3730XL DNA sequencer.

### 2.6. Sequence Analysis

ORF 5 sequences were used for sequence alignment and phylogenetic analysis. The nucleotide sequences of ORF 5 genes were aligned using CLUSTALW [33]. Amino acid sequences were aligned using BioEdit. Nucleotide sequence similarities (as percentages) were assessed. A phylogenetic tree was constructed from the aligned nucleotide sequences using neighbor-joining in MEGA 7 software. Neighbor-joining (NJ) trees were generated with a Kimura 2-parameter model using MEGA 7 [34]. The robustness of the phylogenetic analysis and the significance of the branch order were determined by bootstrap analysis with 1000 replicates.

### 2.7. Antibody Detection

Serum samples were assayed for the presence of PRRSV-specific antibodies by ELISA and SN assays. ELISA (HeardCheck PRRS X3, Idexx Laboratories Inc., Westbrook, ME, USA) was performed in accordance with the manufacturer’s instruction. The presence or absence of antibody was determined by calculating the sample-to-positive control (S/P) ratio of the test. The results were considered positive for PRRSV antibodies when the S/P ratio was greater than 0.4.

An SN assay was performed to titrate PRRSV neutralizing antibodies (NAs). PRRSV NAs were quantified in serum samples from all vaccinated and unvaccinated pigs in MARC-145 cells against the homologous and heterologous viruses. In addition, the immunofluorescence method using a PRRS-hyperimmune serum followed by a fluorescein-labelled anti-porcine IgG antibody (Bio-Rad, Hercules, CA, USA) was used to confirm the NA titers obtained by cytopathogenic effect observation. Neutralization titers were expressed as the reciprocal of the highest serum dilution that completely inhibited virus infection (no CPE).

### 2.8. Analysis of Lymphocyte Cytokine Production of Both INF-γ and IL-10

Heparinized blood was collected at 14 and 35 DPV. PBMCs were isolated by layering the blood on a density gradient medium (LymphoSep™, MP Biomedicals, 9 Goddard, Irvine, CA, USA), followed by gradient centrifugation in accordance with the manufacturer’s instructions. PBMCs were then recalled with homologous virus (vaccine strain), and IFN-γ and IL-10 levels were measured as follows.

#### 2.8.1. Enzyme-Linked Immunospot (ELISPOT) Assay

PBMCs (2 × 10^5^ cells/well) were incubated with either mock media, 0.01 MOI of homologous virus (vaccine strain) or PHA (10 μg/mL, Sigma-Aldrich, St. Louis, MO, USA) for 20 h at 37 °C in 5% CO_2_. IFN-γ production by the treated PBMCs was assessed using an ELISPOT kit (R&D Systems, Minneapolis, MN, USA) according to the manufacturer’s instructions. Spot-forming cells (SFCs) on the PVDF membrane were counted using an ELISPOT Reader (AID ELISPOT Reader, AID GmbH, Strassberg, Germany).

#### 2.8.2. Quantification of IL-10 Levels

IL-10 levels were measured from PBMC supernatants using a porcine ELISA IL-10 kit (R&D Systems, Minneapolis, MN, USA) according to the manufacturer’s protocol. Briefly, PBMCs (2 × 10^6^ cells/well) were cultured with either mock media, 0.01 MOI of homologous virus (vaccine strain) or PHA (10 μg/mL, Sigma-Aldrich, St. Louis, MO, USA) for 20 h at 37 °C in 5% CO_2_. The culture supernatants were collected, and IL-10 levels were quantified by comparison to a porcine IL-10 standard. Optical density (OD) was measured at 450 nm using a microplate reader (Metertech, Taipei, Taiwan).

### 2.9. Data Analysis

All statistical analyses were performed using Statistical Analysis System (SAS) software, version 9.0 (2002, SAS Institute Inc., Cary, NC, USA). All differences in variables between treatment groups were considered significant when *p* < 0.01. To determine the effect of vaccination on growth performance, ADG between weaning and 10 weeks of age was calculated and expressed as the mean ± standard error of mean (mean ± SEM). An analysis of variance was used to compare growth performance between groups (*p* < 0.01). Post hoc pairwise comparisons were then performed using the Tukey test to adjust the *p*-values of these comparisons. The survival rate was calculated, and compared between groups at the end of the experiment. All ELISA values, SN titers, and INF-γ and IL-10 levels are reported as mean ± SEM.

## 3. Results

### 3.1. Mortality and Growth Performance

The survival rates of pigs from the NonVac and US2-MLV groups were 83.36 ± 1.21% and 82.00 ± 0.91%, respectively, while those of pigs in the US1-MLV and US3-MLV groups were 91.32 ± 1.20% and 93.26 ± 1.23%, respectively (Figure 1). The survival rate of pigs in the US3-MLV group was significantly (*p* < 0.001) higher than that of pigs in the other groups (Figure 1).

The body weight gain of pigs was monitored throughout the experiment. As shown in Figure 2, by the end of this study, pigs in the unvaccinated group showed the lowest weight gain and ADG. The pigs in the US3-MLV group had significantly (*p* < 0.001) higher weight gain and ADG (376.10 ± 22.30 g/day) than pigs in the NonVac, US1-MLV, and US2-MLV groups (219.96 ± 21.81, 349.70 ± 21.64 and 322.05 ± 27.76 g/day, respectively) (Figure 2).

### 3.2. Phylogenetic Analysis of PRRSV-2 Isolates

To investigate the strain domination of new Thai PRRSV-2 isolates in the study area, the complete ORF5 genes of 19, 4, and 9 PRRSV-2 isolates collected in 2017, 2019, and 2020, respectively, were analyzed. A phylogenetic tree was constructed for PRRSV-2 isolates. A systematic classification of PRRSV-2 genotype was conducted based on 179 sequences in database including field isolates and vaccine isolates.

PRRSV-2 was divided into nine lineages (1–9) in this system, which was used for PRRSV-2 classification in the present study. Phylogenetic analysis demonstrated that the PRRSV-2 isolates collected in 2017 were classified into only two distinct lineages, with 10 sequences in lineage 8.7/HP and remaining sequences, which appeared to be the novel PRRSV-2 cluster in lineage 1 (Figure 3).

Pairwise nucleotide and amino acid identity values between the clusters collected in 2017 were in the ranges 83.08–84.41% and 83.58–85.07%, respectively (Table 3). Moreover, the 10 sequences in lineage 8.7/HP and 9 sequences in lineage 1 shared 87.56–87.89% and 82.59–83.91% nucleotide sequence identities with Ingelvac^®^ PRRS MLV, respectively. Thai PRRSV-2 isolates in 2019 and 2020 were mostly grouped in the novel cluster classified into lineage 1 in this study herd. Pairwise nucleotide and amino acid identity values of isolates collected in 2017, 2019, and 2020 are summarized in Table 3. Furthermore, the novel isolates shared 82.59–84.42%, 83.75–85.74%, and 84.25–85.90% nucleotides identities with Ingelvac^®^ PRRS MLV, Prime Pac PRRS^®^ MLV, and HP-PRRSV-based vaccine, respectively. The phylogenetic tree demonstrated that lineage 1 Thai PRRSV-2 isolates from 2019–2020 were consistently detected throughout the study and became more dominant in the PRRSV-2 genotype population in the study area over time.

### 3.3. Viremia (RT-PCR) in Serum Samples

PRRSV-1 and PRRSV-2 were not detected in pigs from the US3-MLV group throughout the experiment. In contrast, PRRSV-1 and PRRSV-2 RNA were detected in pigs from the NonVac and US2-MLV groups at 14, 28, 42, and 56 DPV. The pigs in US1-MLV group, PRRSV-1, and PRRSV-2 RNA were detected early at 42 DPV, and PRRSV-1 RNA was detected only at 56 DPV (Figure 4).

### 3.4. Antibody Response as Measured by ELISA

At 0 DPV, PRRSV-specific antibody responses were obviously different in each group of experiment. Pigs in the NonVac and US2-MLV groups showed similar patterns of antibody responses post-vaccination. The antibody responses of pigs in the NonVac and US2-MLV groups showed the earliest seroconversion at 14 DPV and reached a peak between 42 and 56 DPV (Figure 4). At 28 DPV, pigs in the US1-MLV group exhibited the seroconversion.

The antibody response in the NonVac and US2-MLV groups continuously increased and it was significantly (*p* < 0.001) higher than that of US1-MLV and US3-MLV groups (Figure 4). At 42 DPV, seroconversion was detected in the US3-MLV group. The antibody response in the US1-MLV and US3-MLV groups slightly increased and reached a peak at 56 DPV. However, the average antibody levels in the NonVac and US2-MLV groups post-vaccination were significantly (*p* < 0.001) higher than those in the US1-MLV and US3-MLV groups throughout the study (Figure 4).

### 3.5. Antibody Response as Measured by SN Assay

Pigs in the NonVac group remained serologically negative throughout the experiment, as demonstrated by the serum neutralization (SN) assay against homologous virus, which refer to the vaccine isolates shown in Figure 5A. In all vaccinated groups, the SN assay against homologous virus (Figure 5A) showed a similar pattern of SN titers. At 0 DPV, SN titers in the US1-MLV group were significantly (*p* < 0.001) higher than those in the NonVac, US2-MLV, and US3-MLV groups; however, the average SN levels in US2-MLV and US3-MLV groups were not significantly different. In all vaccinated groups, SN levels decreased slightly and reached their lowest level at 14 DPV. Subsequently, SN titers against homologous virus gradually increased, peaking at 42 DPV, with the highest SN titers observed in the US1-MLV group. However, there was no significant difference in SN titers between the US1-MLV and US3-MLV groups post-vaccination. Although the US2-MLV group was one of three vaccinated groups, the SN titer was still lower during the nursery period in the US2-MLV group compared to the US1-MLV and US3-MLV groups (Figure 5A).

In case of the heterologous field PRRSV-2: THA_SP/RB_2783766/S3/17-7, which was collected in 2017 and classified into lineage 8.7/HP (Figure 3), the SN responses against this strain rapidly increased at 14 DPV in the unvaccinated and US-2 MLV vaccinated pigs (Figure 5B). At 28 DPV, the SN titers in pigs from the NonVac and US2-MLV groups were significantly (*p* < 0.001) higher than those from US1-MLV and US3-MLV groups and reached a peak level at 56 DPV. In contrast, the SN responses in the US1-MLV and US3-MLV groups were still low throughout the experiment (Figure 5B). In case of the other heterologous field PRRSV-2: THA_SP/RB_S1/P1/0120-18, which was classified into lineage 1 and shared 83.41% and 84.07% nucleotide and amino acid identities with THA_SP/RB_2783766/S3/17-7, respectively, the SN responses against this strain rapidly increased at 14 DPV in the unvaccinated and US-2 MLV vaccinated pigs (Figure 5C). In contrast, the SN responses in the US1-MLV and US3-MLV groups were still low throughout the experiment (Figure 5C).

### 3.6. Homologous Virus as Vaccine Strain Induced IFN-γ- and IL-10-Producing Lymphocytes

We investigated the ability of the homologous virus (vaccine strain) to activate T lymphocytes by inducing production of IFN-γ and IL-10. The production of these cytokines was analyzed using commercial ELISPOT and ELISA kits. IFN-γ-producing lymphocytes were detected in all vaccination groups except for the NonVac group (Figure 6A). Among the vaccinated groups, the US3-MLV group exhibited the lowest levels of IFN-γ production, while the US2-MLV group showed the highest levels. However, the differences in IFN-γ production among the vaccinated groups were not statistically significant.

Similarly, supernatants from PBMCs stimulated with the homologous virus (vaccine strain) showed the highest IL-10 levels in the US2-MLV group, significantly exceeding those in the other groups, followed by the US1-MLV group (Figure 6B). The US3-MLV group had the significantly lowest IL-10 level. However, the IL-10 levels were not significantly different between the US1-MLV and US2-MLV groups.

## 4. Discussion

The present study was conducted to investigate the field efficacy of MLVs against PRRS in reducing mortality and enhancing growth performances in nursery pigs in a worst case scenario: a single site farrow-to-finish system where different sow and piglet vaccines are used and where the field isolates are genetically dissimilar to the MLV used. We opted to conduct this study in the field instead of a laboratory because this would be the best representation of real life conditions where animals are continuously exposed to different circulating viruses instead of a single controlled challenge.

Growth and mortality rates are the most important zootechnical parameters in evaluating efficacy as PRRSV infection commonly causes decreased growth rates and death from respiratory disease in nursery aged pigs. Based on the obtained results, compared to the unvaccinated control group, both the US3-MLV (Prime Pac PRRS^®^ MLV) and US1-MLV (Ingelvac PRRS^®^ MLV) groups displayed significant reductions in clinical losses and mortality. Compared to that of the unvaccinated pigs, the growth performances of the pigs vaccinated with US3-MLV and US1-MLV were significantly better, as indicated by average daily gain. These results are in line with the previous experiments, which demonstrated that vaccination with PRRSV-MLV significantly increased protection of the progeny pigs against PRRSV infection, as indicated by infection-associated lesions, clinical signs, growth performances, and mortality in a positive herd endemically infected with PRRSV-2 [35,36]. US2-MLV (PRRSV-2 lineage 8 vaccine) was not able to significantly reduce mortality in nursery pigs despite its higher genetic similarity to the field isolates. This suggests our study shows that selection of PRRSV MLV isolates based purely on genetic similarity to field isolates may not yield the best results for producers and veterinarians. Analyzing the results and comparisons between US2-MLV and US3-MLV, we observed differences of 54 g/day and 11% survival rates. This represents a significant economic impact to the end user. Using other publications as a benchmark, for instance, the study by Holtkamp et al. [30] found that in US production conditions, a difference of 10% survival rates and similar reduction in average daily gain cost approximately USD 10.50–12.50 per pig marketed.

One of the predominant wild-type strains circulating in the grow-outs was heterologous PRRSV-2 isolates. A higher SN response was observed in the NonVac group compared to the vaccinated groups. It was speculated that there was greater natural viral exposure on the farm in the NonVac group due to increased shedding of virus, and hence animals were able to mount a greater SN response compared to the MLV groups, where the natural virus exposure may have been lower due to decreased shedding from vaccinated animals. It is noteworthy that, compared to the vaccinated groups and despite mounting a higher SN response, the survival rate and ADG of the NonVac group were clinically inferior. Other authors have also noted that SN responses do not clearly predict clinical outcomes in the case of PRRSV infection [37]. Interestingly, our study found that the US1-MLV and US3-MLV groups were able to elicit better zootechnical performance compared to the US2-MLV group, despite SN responses against the heterologous field strains in US1-MLV and US3-MLV groups being lower. This suggests that the improved performance cannot be attributed purely to humoral immune responses alone. One possible reason for the improved performance may be attributed to cell-mediated immunity (CMI). The role of CMI in controlling PRRSV has been extensively studied in prior publications [38,39,40]. IFN-γ produced by activated T cells and macrophages plays a crucial role in cellular immunity by facilitating inhibited viral replication, antigen presentation, and phagocytosis [41,42]. In the present study, homologous virus stimulation significantly increased IFN-γ-producing lymphocytes in all vaccinated groups compared to the unvaccinated group. This result is aligned with findings from a previous study [43]. The study reported that different commercial MLVs induced CMI slowly, as measured by lymphocyte proliferative response and the number of IFN-γ-PCs, regardless of vaccine genotype. IFN-γ-PCs were detected in pigs at approximately 35 days post-vaccination at low levels; however, the levels increased rapidly after homologous challenge. In prior studies [38,43], challenge by PRRS isolates considered to be heterologous to the vaccine isolates caused unpredictable CMI responses and was found to be unrelated to the genetic similarity between the vaccine and the challenge viruses [43]. With similar CMI induction across all three vaccines, this does not explain the better zootechnical outcome of US1-MLV and US3-MLV groups compared to US2-MLV and the non-vaccinated group.

We suggest that another explanation for the improved zootechnical outcome in US1-MLV and US3-MLV groups may be differences in IL-10 levels induced by vaccination. Elevated IL-10 levels can lead to reinfection or persistence of disease in the host, ultimately hindering PRRSV clearance [44] and potentiating infection. This is supported by findings in other viral infections. In human immunodeficiency virus 1 (HIV-1) infection, IL-10 is produced and is considered an important pathway by which HIV can induce immunodeficiency. In foot-and-mouth disease virus (FMDV) infection, high levels of IL-10 have been demonstrated to mediate immunosuppression and reduced viral clearance in acute FMDV infection. In porcine epidemic diarrhea virus (PEDV) infection, IL-10 is associated with worsened clinical outcomes [45,46]. The immunosuppressive effect of IL-10 has been directly demonstrated in PRRSV infection as well [44]. To support our hypothesis, we assessed the levels induced by the different MLVs in this study. IL-10 levels were measured at 14 DPV, based on a previous study that reported IL-10 levels to be highest at 14 DPV and gradually decreasing by 35 DPV [43]. In our findings, IL-10 levels were detected following homologous in vitro stimulation in all vaccinated groups, with the US2-MLV group showing significantly higher levels than the other vaccinated groups. This suggests that differences exist in IL-10 [47] upregulation between different PRRSV MLV strains and this is possibly a factor explaining the improved performance of US1-MLV and US3-MLV compared to US2-MLV. In our results, the US3-MLV group displayed the lowest levels of IL-10.

The use of different MLVs against PRRS across a single production flow is generally discouraged due to the risks of recombination events. Previous research has indeed shown that recombination events can occur in PRRSV culture-adapted viruses grown in vitro and on farms [26,48,49]. Therefore, best practice would be using a single MLV across an entire production flow to avoid these recombination events and avoid introducing new strains into the herd. In Asia, this is still difficult to implement for various reasons because there are several types of pig production systems, such as one-site farrow-to-finish production systems, two-site systems, and three-site systems. For instance, in three-site systems, a breeding sow herd, nursery, and grow–finish fattening may operate as separate modules and use different MLVs. In the production system chosen, US-1MLV was used for many years in the sow and piglet herd but nursery pigs continued to display clinical signs and mortality associated with PRRSV. Given this, the producer selected to evaluate different vaccines in the piglet herd while keeping US-1MLV in the sow herd. Comparing the performance of US-1MLV and US3-MLV groups, the US3-MLV group gave superior zootechnical performance despite being different from the vaccine used in the sow herd, showing that this approach is a viable approach to PRRSV control. The risk of vaccination recombination can be mitigated by decreased shedding in the field post-vaccination. We have previously shown that the isolate used in US3-MLV has a superior shedding profile vs. US1-MLV [50] and we believe that this is a key factor that should be considered when implementing vaccination programs where one vaccine is used in sows and a separate one is used in piglets.

In this study, the strain domination of PRRSV-2 in the vaccinated herds was investigated using the complete ORF 5 gene of PRRSV-2 isolates collected in 2017, 2019, and 2020. Both PRRSV-1 and PRRSV-2 exist concurrently in the pig population. The phylogenetic analysis demonstrated that PRRSV-2 isolates collected in 2017 were further classified into lineage 8.7/HP, as shown in Figure 3. By 2020, the dominant PRRSV-2 isolates shifted and were mostly grouped in lineage 1 (Figure 3). These sequences shared 82.59–84.42%, 83.75–85.74%, and 84.25–85.90% nucleotide identity with Ingelvac^®^ PRRS MLV, Prime Pac PRRS^®^ MLV, and HP-PRRSV-based vaccines, respectively. Other studies during this period of time have documented that lineage 1 viruses are becoming dominant in the field around the world [51,52], strongly suggesting that this shift in virus dominance may have been due to entry of new viruses into the herd and not due to recombination with MLV isolates used in the farm. Given that PRRS MLV vaccination has been described as “leaky” and does not provide sterilizing immunity [53,54], it is plausible that new field strains continued to filter into the farm during this study. In this situation, we did not find evidence of recombination with either US2-MLV or US3-MLV, despite these being different vaccines from those used in the sow herd. Using different vaccines in the sow and piglet herd is therefore a viable option for producers that have no option to maintain one vaccine throughout the entire production flow. Selecting a vaccine isolate that sheds minimally into the environment will minimize the risk of recombination events occurring on farms that use multiple different MLV isolates.

## 5. Conclusions

Based on overall results of the present study, both US1-MLV and US3-MLV improved production results post-vaccination in piglets, despite decreased genetic similarity to the circulating field virus isolates, with US3-MLV producing the best results. In comparison, another MLV which exhibited closer genetic similarity to the field virus (US2-MLV) displayed lesser improvement compared to US1-MLV and US3-MLV. Separately, the circulating strain on the farm shifted over time from lineage 8.7 to lineage 1, mirroring the shift in PRRSV isolates around the world at this time. The information suggests that recombination between field and vaccine isolates was not occurring, despite the use of different PRRSV MLV isolates in the sow and piglet herds. This suggests that this can be a viable approach in farms that are not able to use a single PRRSV MLV across the entire production system.

Overall, this study supports the statement that genetic similarity between field and vaccine isolates should not be used as a sole criteria to select vaccines. Based on our study findings, we recommend that veterinarians should consider three main aspects when selecting a PRRS MLV. Firstly, the safety profile of the vaccine, specifically the IL-10 levels induced post-vaccination. Secondly, the level of viremia reduction in the field, and finally the zootechnical performance in the field, specifically the reduction in mortality and improvement in average daily gain in the grow-out period.

## 6. Limitations

Some limitations of this study are that we had to work with the production logistics of the producer involved in the study. This meant that we were not able to homogenize pigs by antibody levels at the start of the study and this may have contributed to some confounding effects. As multiple grow-outs were involved, different factors may have influenced the survival rates of pigs, such as difference in staff training and motivation. We attempted to control for this by averaging results across a large number of different nursery grow-out sites. For example, pigs in US1-MLV and US3-MLV groups were weaned at 26 days of age and separated into 56 and 32 different nursery grow-outs, respectively.

## Figures and Tables

**Figure 1 animals-15-00428-f001:**
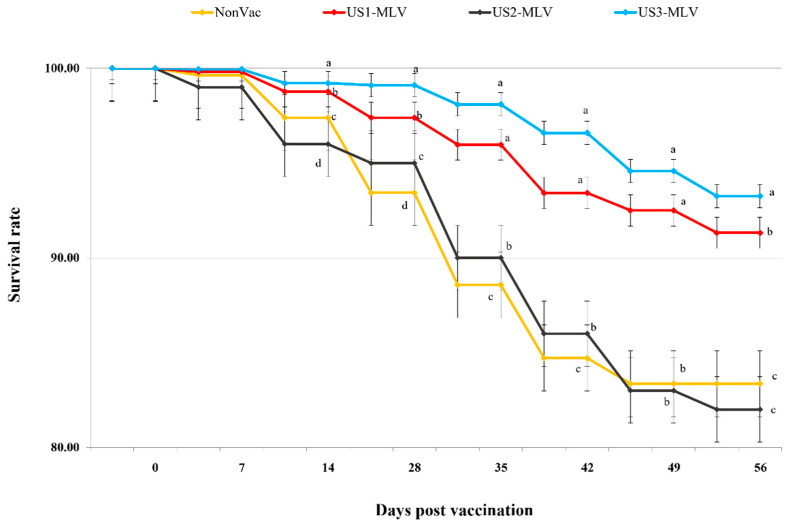
The survival rates in three different PRRSV MLV-vaccinated groups, US1-MLV, US2-MLV, and US3-MLV groups vaccinated with Ingelvac PRRS^®^ MLV, HP-PRRSV-based vaccine and Prime Pac PRRS^®^ MLV, respectively, compared to the non-vaccinated group (NonVac). Variation is expressed as the standard deviation. Different letters in superscript indicate a statistically significant difference (*p*-value < 0.01) between groups.

**Figure 2 animals-15-00428-f002:**
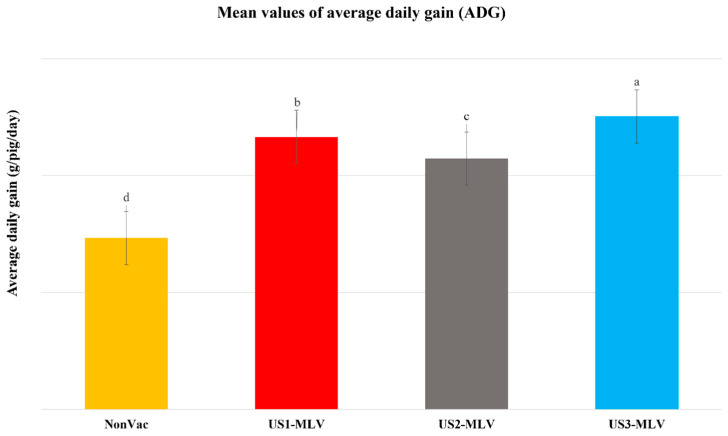
The mean average daily gain (ADG) of four treatment groups (NonVac, US1-MLV, US2-MLV, and US3-MLV) and ADG values are expressed as mean ± standard error of the mean (SEM). Different letters in superscript indicate statistical significant difference (*p*-value < 0.01) between treatment groups.

**Figure 3 animals-15-00428-f003:**
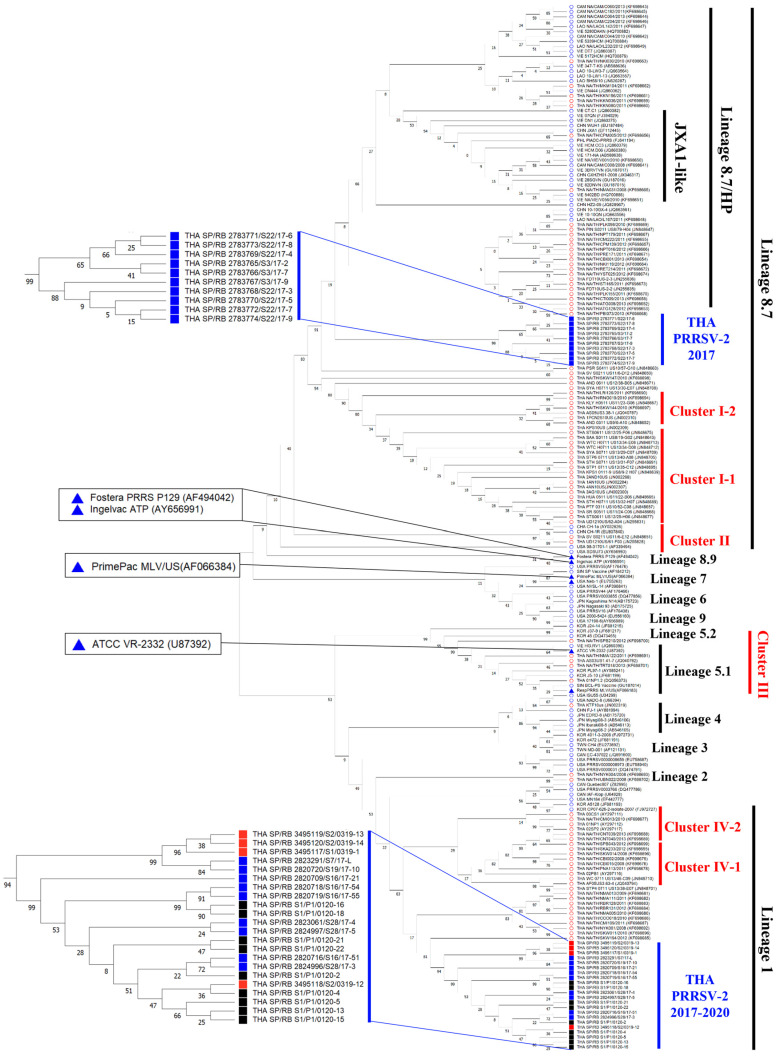
Neighbor-joining tree of PRRSV-2 isolates based on the nucleotide sequences of complete ORF5 genes. Filled triangles represent PRRSV-2 prototype virus (VR-2332) and PRRSV-2 modified live vaccines. The rest of the filled squares represent new Thai PRRSV-2 clusters. The color of filled squares indicates the year of isolation. Blue, red, and black indicate isolation in 2017, 2019, and 2020, respectively.

**Figure 4 animals-15-00428-f004:**
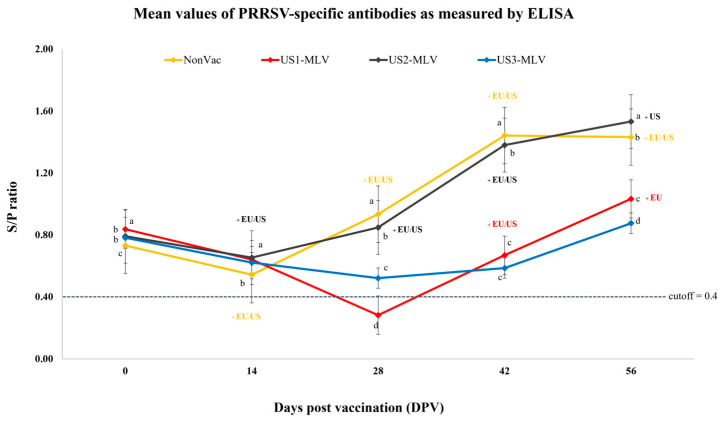
Mean values of PRRSV-specific antibodies as measured by ELISA of four treatment groups, Non-Vac, US1-MLV, US2-MLV, and US3-MLV. Antibody titers are shown as mean ± standard error of the mean (SEM). Different letters within the same DPV indicate statistically significant differences between groups (*p*-value < 0.01). A dashed line indicates the cutoff level (S/P ratio of 0.4). All serum samples collected at 0 day post-vaccination (DPV) were PRRSV ELISA-positive and PCR-negative. Pigs in the US3-MLV group remained ELISA-positive and PCR-negative throughout the study. EU and US indicated positive RNA detection in serum samples by RT-PCR at each DPV.

**Figure 5 animals-15-00428-f005:**
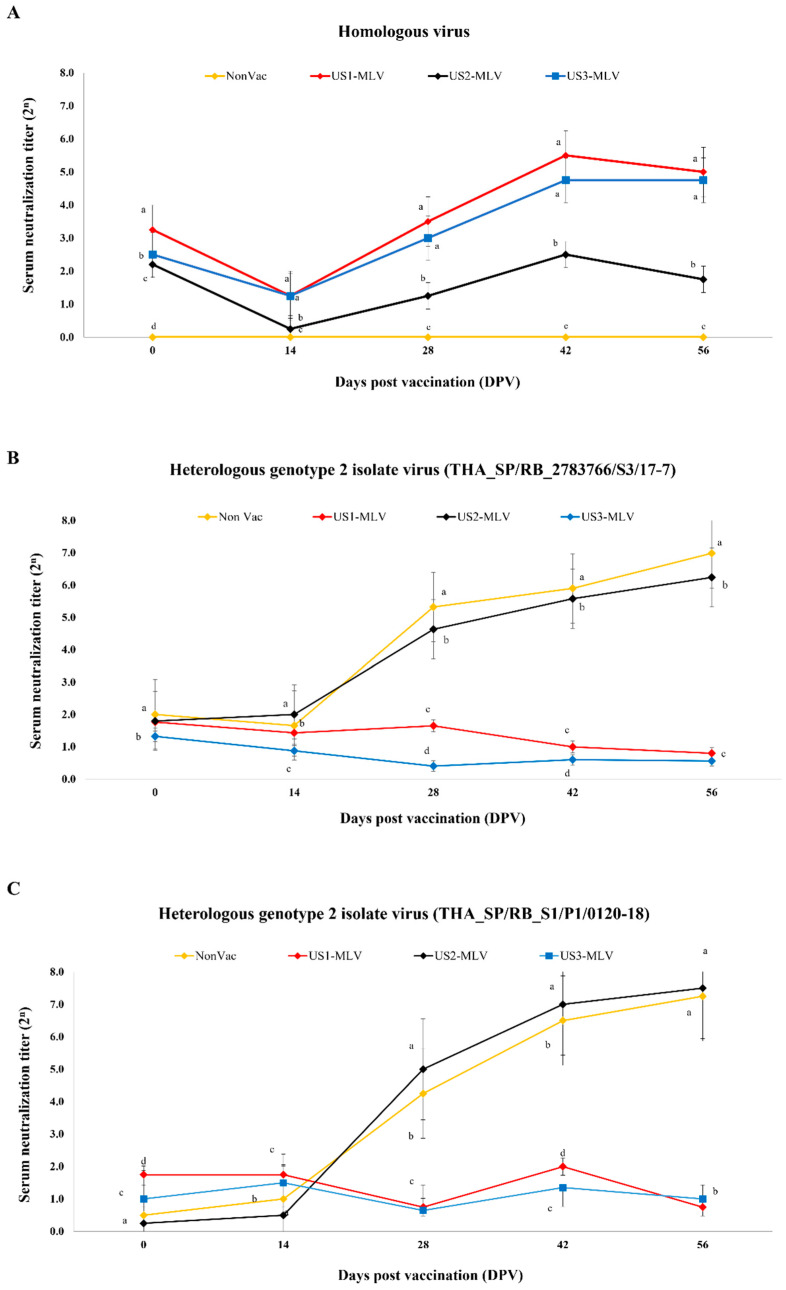
Antibody response measured by the serum neutralizing (SN) assay across four groups: NonVac, US1-MLV, US2-MLV, and US3-MLV. The SN assay was conducted using three different types of viruses: homologous virus (**A**), heterologous PRRSV-2 isolate THA_SP/RB_2783766/S3/17-7 (**B**), and heterologous PRRSV-2 isolate THA_SP/RB_S1/P1/0120-18 (**C**). The SN values are shown as mean ± standard error of the mean (SEM). Different letters within the same DPV indicate statistically significant differences between groups (*p*-value < 0.01).

**Figure 6 animals-15-00428-f006:**
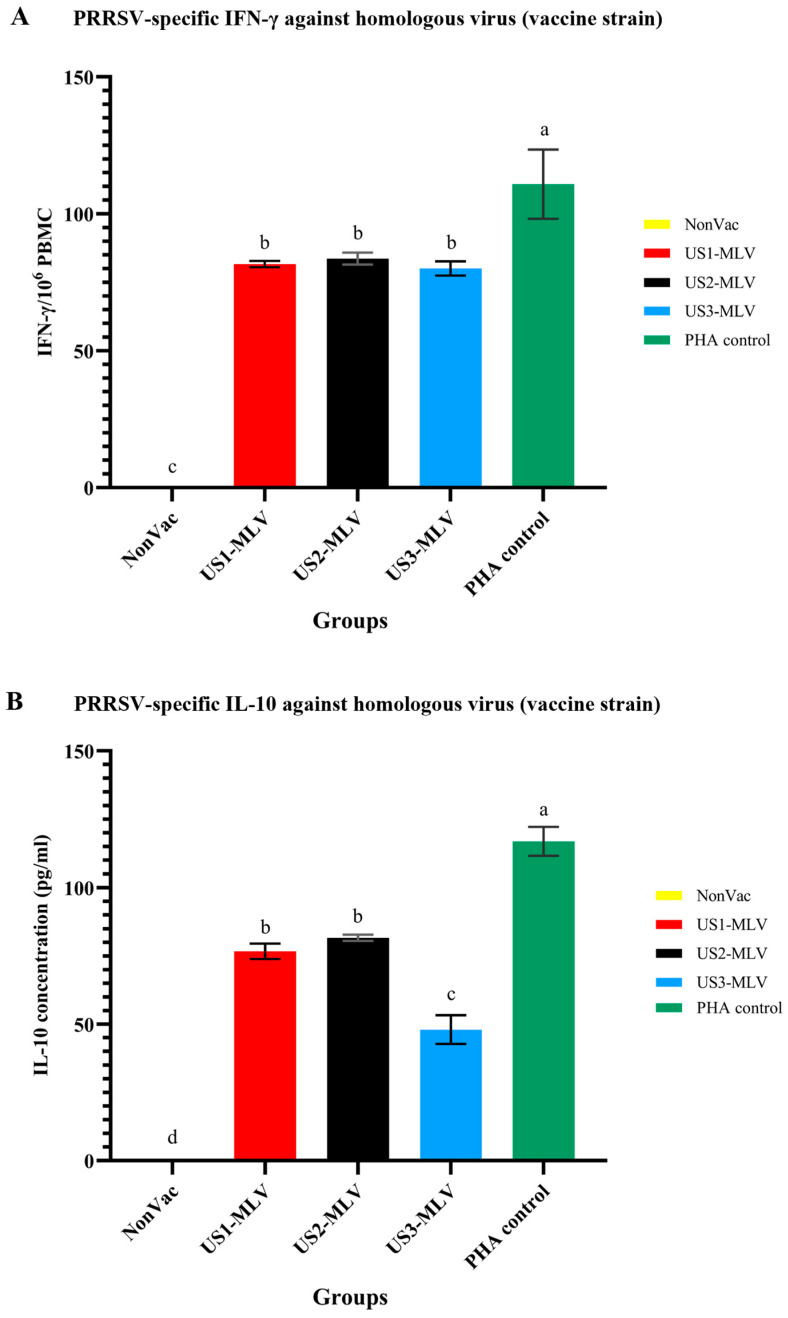
Homologous virus (vaccine strain) and PHA stimulation in vitro assays induced IFN-γ- and IL-10-producing lymphocytes at 35 and 14 DPV, respectively. IFN-γ producing cells (**A**) and IL-10 levels in supernatant of stimulated PBMC cells (**B**) were measured. Data are shown as mean ± standard error of the mean (SEM) from 5 pigs/group. Groups labeled with different letters indicate statistical differences between groups (*p*-value < 0.01).

**Table 1 animals-15-00428-t001:** Experimental design. The pigs were allocated into 4 treatment groups and vaccinated with 3 different PRRS MLVs. The NonVac group was included as the unvaccinated control group.

Treatment Groups	No. of Pigs	Vaccination	Vaccine	Dosage and Route of Administration	Manufacturers
NonVac	1112	No	-	-	-
US1-MLV	51,535	Yes	Ingelvac^®^ PRRS MLV	2 mL, intramuscular	Boehringer Ingelheim, Ingelheim am Rhein, German
US2-MLV	1200	Yes	HuN4-F112 PRRS MLV	2 mL, intramuscular	Harbin Veterinary Research Institute, CAAS, Harbin, China
US3-MLV	22,228	Yes	PrimePac^®^ PRRS MLV	1 mL, intramuscular	MSD Animal Health, Rahway, NJ, USA

**Table 2 animals-15-00428-t002:** Nucleotide and amino acid identities based on ORF5 gene from vaccine strains and field PRRSV-2 isolates.

PRRSV-2 Isolate	Nucleotide and Amino Acid Identities			
	Level of Similarity	HP-PRRSV Based Vaccine	Ingelvac^®^ PRRS MLV	Prime Pac^®^ PRRS MLV
THA_SP/RB_S1/P1/0120-18	Nucleotide	84.1%	82.1%	83.1%
	Amino acid	83.5%	80.0%	84.0%
THA_SP/RB_2783766/S3/17-7	Nucleotide	91.4%	87.6%	88.1%
	Amino acid	89.1%	85.1%	88.6%

**Table 3 animals-15-00428-t003:** The percentage identity of the nucleotide and amino acid sequences of the ORF5 gene in the Thai PRRSV-2 isolates collected in 2017 to 2020. Italic font indicates nucleotide identities, and bold font indicates amino acid identities.

Cluster	2017 (Lineage 8.7/HP)	2017 (Lineage 1)	2019	2020
2017 (lineage 8.7/HP)	99.67–100.00% 99.50–100.00%	83.08–84.41% 83.58–85.07%	84.42–83.25% 83.58–85.07%	83.08–84.08% 84.07–85.07%
2017 (lineage 1)	–	91.71–99.67% 91.04–99.00%	90.22–99.00% 89.05–99.00%	98.18–99.67% 97.01–100.00%
2019		–	90.88–100.00% 91.05–100.00%	90.88–91.38% 89.95–91.05%
2020			–	98.67–99.83% 98.01–100.00%

## Data Availability

The data that support the findings of this study are available on request from the corresponding author.

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
