# Peer review of "Field Investigation Evaluating the Efficacy of Porcine Reproductive and Respiratory Syndrome Virus Type 2 (PRRSV-2) Modified Live Vaccines in Nursery Pigs Exposed to Multiple Heterologous PRRSV Strains"

_animals, 2025, doi:10.3390/ani15030428_

Round 1
Reviewer 1 Report (Previous Reviewer 2)
Comments and Suggestions for Authors
1. The sequence numbers of Lines in Response are completely inconsistent with those in the manuscript. The changes are also not highlighted in blue.
2. The discussion about IL-10 detection results in the text (Lines 672-677 in "Animal-3419806-peer-review-V1") and the relevant part in the response document (Revised manuscript Line 614-631) is inconsistent, which is accurate?
3. The authors added the detection results of IFN-γ (35 dpv) and IL-10 (14 dpv), but in 2.8.1, the authors set media and PHA controls, and in 2.8.2, the authors set PHA control, which was set for PBMCs from NonVac and all three immunization groups, or just for one group? No control results are presented in Figure 6A and Figure 6B.

Author Response
Point-by-point response to reviewer 1
Reviewer 1 comments
Comment 1: The sequence numbers of Lines in Response are completely inconsistent with those in the manuscript. The changes are also not highlighted in blue.
Response to Reviewer 1 for the comment 1
Thank you for your kind suggestions and comments. First, we apologize for any misunderstanding. The sentences highlighted in blue represent the second-round revision made prior to the rejection. Subsequently, we revised and resubmitted the manuscript, removing the blue highlights and using green highlights to indicate responses to the Editor's comments. In the current version, the revised sentences are highlighted in grey.
Comment 2: The discussion about IL-10 detection results in the text (Lines 672-677 in "Animal-3419806-peer-review-V1") and the relevant part in the response document (Revised manuscript Line 614-631) is inconsistent, which is accurate?
Response to Reviewer 1 for the comment 2
Again, as comment 1, we apologize for the misunderstanding. The discussion regarding the IL-10 detection results was presented in Lines 614-631 of the second-round revision (prior to rejection) and in Lines 664-677 of "Animal-3419806-peer-review-V1". Subsequently, we revised this section and resubmitted the manuscript. Currently, the updated discussion can be found in Lines 667-697 of this revised manuscript.
Revised manuscript Lines 667-697:
We suggest that another explanation for the improved zootechnical outcome in US1-MLV and US3-MLV may be differences in IL-10 levels induced by vaccination. Elevated IL-10 levels can lead to reinfection or persistence of disease in the host, ultimately hindering PRRSV clearance33 and potentiating infection. This is supported by findings in other viral infections. In human immunodeficiency virus 1 (HIV-1) infection, IL-10 is produced and is considered an important pathway by which HIV can induce immunodeficiency. In foot-and-mouth disease virus (FMDV) infection, high levels of IL-10 have been demonstrated to mediate immunosuppression and reduced viral clearance in acute FMDV infection. In porcine epidemic diarrhea virus (PEDV) infection, IL-10 is associated with worsened clinical outcomes34.The immunosuppresive effect of IL-10 has been directly demonstrated in PRRSV infection as well33. To support our hypothesis, we assessed the levels induced by the different MLVs in this study. IL-10 levels were measured at 14 DPV, based on a previous study that reported IL-10 levels to be highest at 14 DPV and gradually decreasing by 35 DPV32. In our findings, IL-10 levels were detected following homologous in vitro stimulation in all vaccinated groups, with the US2-MLV group showing significantly higher levels than the other vaccinated groups. This suggests that differences exist in IL-10 36 upregulation between different PRRSV MLV strains and is possibly a factor explaining the improved performance of US1-MLV and US3-MLV compared to US2-MLV. In our results, the US3-MLV group displayed the lowest levels of IL-10.
Comment 3: The authors added the detection results of IFN-γ (35 dpv) and IL-10 (14 dpv), but in 2.8.1, the authors set media and PHA controls, and in 2.8.2, the authors set PHA control, which was set for PBMCs from NonVac and all three immunization groups, or just for one group? No control results are presented in Figure 6A and Figure 6B.
Response to Reviewer 1 for the comment 3
Thank you for your comment. We would like to clarify that all samples in all groups were stimulated with PHA as positive control in vitro test. In addition, to enhance understanding, we have added the term "mock media," as mentioned in line 365 and highlighted in grey.
Revised manuscript Lines 364-366: Briefly, PBMC (2 × 106 cells/well) were culture with either mock media, 0.01 MOI of homologous virus (vaccine strain) or PHA (10 μg/ml, Sigma-Aldrich, St. Louis, MO, USA)
Figures 6A and 6B illustrate all groups included in this in vivo experiment. We believe it is unnecessary to include PHA in the graphs as a control group.v
Reviewer 2 Report (Previous Reviewer 3)
Comments and Suggestions for Authors
The aim of the article ‘’Field investigation evaluating the efficacy of porcine reproductive and respiratory syndrome virus type 2 (PRRSV-2) modified live vaccines in nursery pigs exposed to multiple heterologous PRRSV strains” was to understand the improvement in clinical parameters in piglets where different modified live vaccines (MLVs) against porcine reproductive and respiratory syndrome (PRRS) were administered. This is a well-designed and well-written work, providing interesting results and novel results. With revisions to improve clarity, data presentation and interpretation, the manuscript will provide valuable insights to optimize PRRS vaccination strategies. Therefore, I suggest the acceptance under a major revision based on the following comments.
Comments for the authors
Introduction
- The introduction is well structured but could benefit from more emphasis on the challenges of genetic diversity in PRRSV and its impact on vaccine efficacy.
Materials and Methods
- Explain the reasons for the selection of the specific MLV vaccines used. Were they selected based on previous genetic analysis or market availability?
- Indicate how the pigs were randomly selected for blood sampling to ensure unbiased representation.
Discussion:
- Provide more details about the discrepancy between genetic similarity and vaccine performance. Is it possible that different immune mechanisms explain why the US3-MLV performs better than the US2-MLV because it has different immune mechanisms? If this is the case, please explain these mechanisms.
Conclusions
- The conclusion section is well structured but would be strengthened by a recommendation on the criteria for vaccine selection.
Author Response
Point-by-point response to reviewer 2
Reviewer 2 comments
The aim of the article ‘’Field investigation evaluating the efficacy of porcine reproductive and respiratory syndrome virus type 2 (PRRSV-2) modified live vaccines in nursery pigs exposed to multiple heterologous PRRSV strains” was to understand the improvement in clinical parameters in piglets where different modified live vaccines (MLVs) against porcine reproductive and respiratory syndrome (PRRS) were administered. This is a well-designed and well-written work, providing interesting results and novel results. With revisions to improve clarity, data presentation and interpretation, the manuscript will provide valuable insights to optimize PRRS vaccination strategies. Therefore, I suggest the acceptance under a major revision based on the following comments.
Comments for the authors
Introduction
- The introduction is well structured but could benefit from more emphasis on the challenges of genetic diversity in PRRSV and its impact on vaccine efficacy.
Response to Reviewer 2 for the introduction part
Thank you for your kind comments. PRRS is indeed a highly genetically variable virus and commercial MLV vaccines have difficulty keeping up with the changes in virus in the field. This means that producers struggle to match vaccine strains to field strains and a different approach is needed. We have added additional information on the challenges of matching MLV isolates from commercial vaccines and field strains into the Introduction.
Revised manuscript Lines 110-131:
Firstly, the coexistence of PRRSV-1 and PRRSV-2 infection has been increasingly reported in Asia11-13. The presence of co-infection of PRRSV-1 and PRRSV-2 in farms complicates the selection of MLV vaccines, as the ORF5 sequences from these two genomes are highly dissimilar. Secondly, multiple PRRS MLV are commercially available in Asia and based on strains such as VR2332, NEB-1, HuN4-F112, JXA1-P80, CH-1R or R98. The strains VR2332, CH-1R and R98 are of lineage 545, NEB-1 is lineage 746 and the others belong to lineage 847. The high mutation rate of PRRS means that new isolates of PRRS are constantly emerging in the field through mutation or recombination. PRRSV has one of the highest mutation rates of RNA viruses. Different studies have suggested a mutation rate between 4.71 × 102 to 9.8 × 102/synonymous sites/year49 52 53 .Recombination is also frequently seen in the field with PRRS isolates, either between field strains or between field and vaccine strains, often resulting in new PRRS genotypes emerging, sometimes with increased virulence50, 51. Regulatory approval of new PRRS MLV vaccines can take years before commercial licences are granted48. Consequently, this means that producers are forced to use existing commercially available vaccines to provide cross protection against emerging PRRS field strains and there is hence need to evaluate existing vaccines on their ability to provide cross protection against contemporary circulating PRRS field strains.
Materials and Methods
Comment 1: Explain the reasons for the selection of the specific MLV vaccines used. Were they selected based on previous genetic analysis or market availability?
Response to Reviewer 2 for comment 1 in the materials and methods part
We have clarified the reasons for the selection of the MLV vaccines. Essentially, they were the ones available to the farm at the time and were in their farm inventory and available to be used in the study.
Revised Manuscript Lines: 213-216
US1-MLV, US2-MLV, and US3-MLV were selected as these vaccines were commercially available to the company operating the farms and were in inventory at the time of the study and hence available to be used by the farms.
Comment 2: Indicate how the pigs were randomly selected for blood sampling to ensure unbiased representation.
Response to Reviewer 2 for comment 2 in the materials and methods part
Thank you for your suggestion. Within the confines of the production system, we did our best to ensure unbiased representation by the following method below. We have clarified this within the text.
Revised Manuscript Lines 230-244:
The assignment of pigs into the nursery grow-outs was conducted by the farm management and the study monitors were not able to control this. For US2-MLV and NonVac groups, there were only 2 grow outs available to use so all were used for blood collection. For the US1-MLV and US3-MLV groups, two grow outs were selected by random selection for blood collection. Each grow-out was plotted onto a spreadsheet (Microsoft Excel®) and a 20 sided dice was rolled. Only if the dice came up to 1, would the site be selected. If no sites managed to roll a 1, unselected sites would be assigned again until 2 sites were found. For the selection of pigs within the selected sites, each site would be filled with approximately 600 pigs. On arrival, pigs were divided into 30 pens of 20 pigs as they came off the transport. Study monitors would roll the 20 sided dice for every pen until a 1 was found. If a 1 was found, a pig would be selected within that pen to be ear tagged for blood collection until 5 pigs were identified.
Discussion:
- Provide more details about the discrepancy between genetic similarity and vaccine performance. Is it possible that different immune mechanisms explain why the US3-MLV performs better than the US2-MLV because it has different immune mechanisms? If this is the case, please explain these mechanisms.
Response to Reviewer 2 for the discussion part
Thank you for your suggestion. We postulate that differences in IL-10 induction accounted for the differences in zootechnical outcomes of US3-MLV vs US2-MLV. We have rewritten this segment below to clarify our meaning.
Revised Manuscript Lines 667-697:
We suggest that another explanation for the improved zootechnical outcome in US1-MLV and US3-MLV may be differences in IL-10 levels induced by vaccination. Elevated IL-10 levels can lead to reinfection or persistence of disease in the host, ultimately hindering PRRSV clearance33 and potentiating infection. This is supported by findings in other viral infections. In human immunodeficiency virus 1 (HIV-1) infection, IL-10 is produced and is considered an important pathway by which HIV can induce immunodeficiency. In foot-and-mouth disease virus (FMDV) infection, high levels of IL-10 have been demonstrated to mediate immunosuppression and reduced viral clearance in acute FMDV infection. In porcine epidemic diarrhea virus (PEDV) infection, IL-10 is associated with worsened clinical outcomes34. The immunosuppresive effect of IL-10 has been directly demonstrated in PRRSV infection as well33. To support our hypothesis, we assessed the levels induced by the different MLVs in this study. IL-10 levels were measured at 14 DPV, based on a previous study that reported IL-10 levels to be highest at 14 DPV and gradually decreasing by 35 DPV32. In our findings, IL-10 levels were detected following homologous in vitro stimulation in all vaccinated groups, with the US2-MLV group showing significantly higher levels than the other vaccinated groups. This suggests that differences exist in IL-10 36 upregulation between different PRRSV MLV strains and is possibly a factor explaining the improved performance of US1-MLV and US3-MLV compared to US2-MLV. In our results, the US3-MLV group displayed the lowest levels of IL-10.
Conclusion
- The conclusion section is well structured but would be strengthened by a recommendation on the criteria for vaccine selection.
Response to Reviewer 2 for the conclusion part
Thank you for your suggestion. We have clarified our suggested criteria for vaccine selection, based on our findings.
Revised Manuscript Lines 771-778:
Based on our study findings, we recommend that veterinarians should consider three main aspects when selecting a PRRS MLV. Firstly, the safety profile of the vaccine, specifically the IL-10 levels induced post vaccination. Secondly, the level of viremia reduction in the field and finally the zootechnical performance in the field, specifically the reduction in mortality and improvement in average daily gain in the grow-out period.
Round 2
Reviewer 1 Report (Previous Reviewer 2)
Comments and Suggestions for Authors
The authors isolated and stimulated the experimental pig PBMC in vitro, and then used ELISPOT or ELISA to detect cytokine levels. Why do they think it is unnecessary to include PHA in the graphs as a control group? Without a PHA control group, how does the effectiveness of the test reflect?

Author Response
Point-by-point response to reviewer 1
Comments and Suggestions for Authors
The authors isolated and stimulated the experimental pig PBMC in vitro, and then used ELISPOT or ELISA to detect cytokine levels. Why do they think it is unnecessary to include PHA in the graphs as a control group? Without a PHA control group, how does the effectiveness of the test reflect?
Response to Reviewer 1
Thank you for your kind comments and suggestions. We apologize for our mistakes and misunderstandings. In response to your feedback, we have edited and added graph for the PHA control group in both the IFN-γ and IL-10 results to improve the reliability of the experiment. Additionally, we have updated the figure legend, which can be found in Lines 589-590 of the revised manuscript, highlighted in grey.
Revised manuscript Lines 589-590: Figure 6. Homologous virus (vaccine strain) and PHA stimulation in vitro assays induced IFN-γ- and IL-10 producing lymphocytes

This manuscript is a resubmission of an earlier submission. The following is a list of the peer review reports and author responses from that submission.
Round 1
Reviewer 1 Report
Comments and Suggestions for Authors
The manuscript by Mebumroong et al described the efficacy of PRRSV-2 MLV in nursery pigs exposed to multiple heterologous PRRSV strains. This study contains the valuable data for application of PRRSV MLV. However, I have several concerns as follows:
Major
1, Although the data of survival rate looks beautiful, in fact, it is not so persuaded. There are several factors which influence the survival rates, including the different numbers of pigs among those groups, reasons of death and so on. How would the authors understand this?
2, How to understand only 5 pigs randomly chosen for analysis in each group? Could those 5 pigs be representative of the whole group?
3, How to understand the higher SN titter to heterologous genotype 2 in non Vac group compared to the MLV groups?
Minor
1, How to understand the letters in superscript in each time point for survival rates?
Author Response
Comment 1: The manuscript by Mebumroong et al described the efficacy of PRRSV-2 MLV in nursery pigs exposed to multiple heterologous PRRSV strains. This study contains the valuable data for application of PRRSV MLV. However, I have several concerns as follows:
Major
1, Although the data of survival rate looks beautiful, in fact, it is not so persuaded. There are several factors which influence the survival rates, including the different numbers of pigs among those groups, reasons of death and so on. How would the authors understand this?
2, How to understand only 5 pigs randomly chosen for analysis in each group? Could those 5 pigs be representative of the whole group?
3, How to understand the higher SN titter to heterologous genotype 2 in non Vac group compared to the MLV groups?
Response 1:
Thank you for your comments and thoughtful insights. We provide a point by point response below to your kind comments, divided by Major and Minor.
For major comments
- We agree that in swine production, different factors may have influenced the survival rates of pigs. For example, difference in staff training and quality for example may have played a part in the differing survival rates. We acknowledge that this was a limitation of the trial as we had to work with the logistics of the company we were working with. To mitigate this, we attempted to average out survival information across grow-out sites. We have acknowledged this limitation in our Conclusion as well.
Revised manuscript Line 607 – 618:
- Limitations
Some limitations of this study are that we had to work with the logistics and delivery schedules of the company involved in the study. This meant that we were not able to homogenize pigs by antibody levels at the start of the study and this may have contributed to some confounding effects. As multiple grow-outs were involved, different factors may have influenced the survival rates of pigs, such as difference in staff training and motivation. We attempted to control for this by averaging results across different nursery grow-out sites. For example, Pigs in US1-MLV and US3-MLV were weaned at 26 days of age and separated into 56 and 32 different nursery grow-outs, respectively.
- Due to the logistics of the study, we were only able to follow 5 animals per site and a total of 2 sites per group (Total of 10 animals) for continuous blood collection. We ear-tagged these 10 animals and collected blood only from them at each timepoint. We believe that this would truthfully show the evolution of viremia and PRRSV detection across 0,14,28,42 and 56 DPV. In terms of survival analysis, this was taken from company records and reflects the true survival status of the entire group (e.g 51,535 pigs vaccinated with US1-MLV). We have clarified this in the materials and methods section of the publication.
Revised manuscript Line 223-226: Survival rate and average daily gain (ADG) were collected from company data records across all animals and grow-outs enrolled in the study. The person responsible for assessing the mortality, survival rate, feed intake and ADG was the farm's supervising veterinarian.
- We believe that for the Non-Vac group, the predominant wild type strain circulating in these grow-outs was Heterologous Genotype 2 isolate virus. It is possible that in the Non Vac groups, there was greater viral exposure on farm and hence animals were able to mount a greater SN response vs the MLV vaccines where the vaccine strains may not have fully matched the challenge strain. However we note that compared to the vaccinated groups and despite mounting a higher SN response, the survival and ADG rates of the Non-Vac group were clinically inferior. We have also added this into the Discussion section.
Revised manuscript Line 525-534: One of the predominant wild type strains circulating in the grow-outs was Heterologous Genotype 2 isolate virus. It is possible that in the Non Vac groups, there was greater natural viral exposure on farm due to increased shedding of virus and hence animals were able to mount a greater SN response vs the MLV groups where the natural virus exposure may have been lower due to decreased shedding from vaccinated animals. We note that compared to the vaccinated groups and despite mounting a higher SN response, the survival and ADG rates of the Non-Vac group were clinically inferior.
Minor
1, How to understand the letters in superscript in each time point for survival rates?
For minor comments
For better understanding. We compared the survival rates between groups at each time point. The superscript letters 'a', 'b', 'c', and 'd' indicate the levels of significant differences among the groups at that time point, with 'a' representing the highest difference and 'd' the fourth high.
For example, at 14 DPV, “a” indicates US3-MLV which has the highest survival rate and shows significant differences compared to US1-MLV, NonVac, and US2-MLV. “b” indicates US1-MLV, which has the second high survival rate and shows significant differences compared to US3-MLV, NonVac, and US2-MLV. “c” indicates NonVac, which has the third high survival rate and shows significant differences compared to US1-MLV, US2-MLV, and US3-MLV. “d” indicates US2-MLV, which has the fourth high survival rate and shows significant differences compared to US1-MLV and US3-MLV.

Reviewer 2 Report
Comments and Suggestions for Authors
In this study, by comparing the specific antibody, serum neutralization (SN), viremia, average daily gain (ADG), and survival rate, the protective efficacy of 3 kinds of porcine reproductive and respiratory syndrome virus 2 (PRRSV-2) modified live vaccines (MLV) in nursery pigs in the worst case scenario where MLV does not match the genetic profile of the field isolate were evaluated. The authors stated that the results of this study support the notion that genetic similarity between field and vaccine isolates should not be used as a sole criterion for selecting vaccines. Zootechnical performance and viremia reduction should also be the final indicators of vaccine selection, coupled with the vaccine safety profile.
Strength:
The results provide the theoretical basis and data reference for the study herd to select vaccines.
Weakness:
1. The authors note that the improved performance cannot be attributed purely to humoral immune responses alone. A possible explanation is the effect of cell-mediated immunity (CMI). Why did the authors not consider testing CMI when designing the trial?
2. The author also suggests that IL-10 has been documented to have an immunosuppressive effect. This may also account for the improved performance of US1-MLV and US3-MLV, despite the decreased humoral response vs US2-MLV. Why did the authors design the trial without considering the detection of IL-10 levels?
3. Did the authors measure antibody levels before grouping the piglets? At the beginning of the test, there was a significant difference in the PRRSV-specific antibody level and the serum-neutralizing antibody among the groups. Did it affect the test result? Especially in the US1-MLV group, the level of specific antibodies at the beginning of the trial is higher, will it affect the immune effect of the vaccine? Will it cause the antibody level of the test pigs to decline or even turn negative after immunization? Serum-neutralizing antibody levels against homologous strains in the US1-MLV group were significantly lower than those in the other two groups. Did that make a difference?
Specific points:
1. Is there any relevant information for the vaccine “PRRSV lineage 8.7 based MLV”? Who made the vaccine? How is it prepared? How to determine the effectiveness? What is the source and other information of the vaccine strain?
2. Change “Table 1” to three-line formï¼›
3. In table 2,the data of THA_SP/RB_2783766/S3/17-7 was missed?
4. In line 317, “was detected only at 56 DPV (Fig.4)”. But Fig. 4 is the results of the PRRSV-specific antibody.
5. In lines 320-321, “……the complete ORF5 genes of PRRSV-2 isolate collected in 2017, 2019 and 2020 were analyzed.” The number of isolates each year and the names of the isolates should be clearly stated.

Author Response
Comment 1: In this study, by comparing the specific antibody, serum neutralization (SN), viremia, average daily gain (ADG), and survival rate, the protective efficacy of 3 kinds of porcine reproductive and respiratory syndrome virus 2 (PRRSV-2) modified live vaccines (MLV) in nursery pigs in the worst case scenario where MLV does not match the genetic profile of the field isolate were evaluated. The authors stated that the results of this study support the notion that genetic similarity between field and vaccine isolates should not be used as a sole criterion for selecting vaccines. Zootechnical performance and viremia reduction should also be the final indicators of vaccine selection, coupled with the vaccine safety profile.
Strength:
The results provide the theoretical basis and data reference for the study herd to select vaccines.
Weakness:
- The authors note that the improved performance cannot be attributed purely to humoral immune responses alone. A possible explanation is the effect of cell-mediated immunity (CMI). Why did the authors not consider testing CMI when designing the trial?
- The author also suggests that IL-10 has been documented to have an immunosuppressive effect. This may also account for the improved performance of US1-MLV and US3-MLV, despite the decreased humoral response vs US2-MLV. Why did the authors design the trial without considering the detection of IL-10 levels?
- Did the authors measure antibody levels before grouping the piglets? At the beginning of the test, there was a significant difference in the PRRSV-specific antibody level and the serum-neutralizing antibody among the groups. Did it affect the test result? Especially in the US1-MLV group, the level of specific antibodies at the beginning of the trial is higher, will it affect the immune effect of the vaccine? Will it cause the antibody level of the test pigs to decline or even turn negative after immunization? Serum-neutralizing antibody levels against homologous strains in the US1-MLV group were significantly lower than those in the other two groups. Did that make a difference?
Response 1:
Thank you for your comments and thoughtful insights. We provide a point by point response below to your kind comments, grouped by Weakness and Specific Points.
Response for weakness points
Points 1 and 2: We acknowledge that CMI and IL-10 are potential points that could have been measured. This may also help to explain the different ADG and survivability of the groups. We will factor this into future trial designs.
Points 3: We were not able to homogenize pigs by antibody levels due to the production requirements of the company. In this instance, we acknowledge that this is a limitation that could have confounded the results and we have acknowledged this in the conclusion as well.
Revised manuscript Line 608 – 612: Some limitations of this study are that we had to work with the logistics and delivery schedules of the company involved in the study. This meant that we were not able to homogenize pigs by antibody levels at the start of the study and this may have contributed to some confounding effects.
Specific points:
- Is there any relevant information for the vaccine “PRRSV lineage 8.7 based MLV”? Who made the vaccine? How is it prepared? How to determine the effectiveness? What is the source and other information of the vaccine strain?
- Change “Table 1” to three-line formï¼›
- In table 2,the data of THA_SP/RB_2783766/S3/17-7 was missed?
- In line 317, “was detected only at 56 DPV (Fig.4)”. But Fig. 4 is the results of the PRRSV-specific antibody.
- In lines 320-321, “……the complete ORF5 genes of PRRSV-2 isolate collected in 2017, 2019 and 2020 were analyzed.” The number of isolates each year and the names of the isolates should be clearly stated.
Response for specific points
- We agree and have added additional information in the revised manuscript. This vaccine is a 2-ml dose of PRRS MLV (Harbin Veterinary Research Institute, CAAS, China) It is a live vaccine containing PRRSV- 2 lineage 8.7 HuN4-F112 strain. Additionally, we change the name of the vaccine in Table 1 from “HP-PRRSV-based vaccine” to “HuN4-F112 PRRS MLV” in the US2-MLV group.
Revised manuscript Line 207-209: “a 2-ml dose of HuN4-F112 (Harbin Veterinary Research Institute, CAAS, China), a PRRSV-2 lineage 8.7 based MLV”
- Thank you for your suggestion. “Table 1” has been amended to three-line form as presented between Line 229-230 in the revised manuscript.
- We agree. The data of THA_SP/RB_2783766/S3/17-7 was added to table 2 in a three-line form, similar to Table 1.
- We agree and modified Figure 4 and its legend. We revised Figure 4 by adding labels for the EU and US to indicate positive RT-PCR results for detecting RNA of PRRSV in serum samples. The Figure 4 legend now explains the RT-PCR results, as detailed in the revised manuscript on lines 420-421.
Revised manuscript Line 420-421: EU and US indicated positive RNA detection in serum samples by RT-PCR at each DPV.
- Thank you for your suggestions, I have already added the number of new Thai isolates for each year as presented in line 353-355 of the revised manuscript. The names of the new Thai isolates for each year are identified by different colors as described in the figure 3 legend.
Revised manuscript Line 353-355: To investigate the strain domination of new Thai PRRSV-2 isolates in the study area, the complete ORF5 genes of 19, 4, and 9 PRRSV-2 isolates collected in 2017, 2019, and 2020, respectively, were analyzed.

Reviewer 3 Report
Comments and Suggestions for Authors
The aim of the article ‘’Field investigation evaluating the efficacy of porcine reproductive and respiratory syndrome virus type 2 (PRRSV-2) modified live vaccines in nursery pigs exposed to multiple heterologous PRRSV strains” was to evaluate the protective efficacy of porcine reproductive and respiratory syndrome virus 2 (PRRSV-2) modified live vaccines (MLV) in nursery pigs in a worst-case scenario where MLV does not match the genetic profile of the field isolate. This study deals with an interesting topic and fits with the journal's scope. This is a well-designed and well-written work, providing interesting results and novel results. Therefore, I suggest the acceptance under a major revision based on the following major and minor comments.
Comments for the authors
Abstract
- Please state clearly the aim of your study.
- Please summarize the information presented in the abstract. The text is over 200 words long.
- In the keywords place nursery pigs in the first place, it is the focus species of the study.
Materials and Methods
- Provide details of the vaccination program of the trial piglets.
- Provide more data on whether the piglets received any preventive treatment.
- Provide further details on the method of blood collection (needle size, venipuncture), the method of obtaining serum from the blood samples and who was responsible for the assessment of mortality, feed intake, survival rate and ADG.
- Has there been any sample size calculation?
Discussion:
- You could discuss the economic impact of your results on pig production.
- What are the negative points of the research, if any? Your recommendations for future studies, if any?
Comments on the Quality of English Language- Moderate editing regarding English language is required for the improvement of the text.
Author Response
Comment 1: The aim of the article ‘’Field investigation evaluating the efficacy of porcine reproductive and respiratory syndrome virus type 2 (PRRSV-2) modified live vaccines in nursery pigs exposed to multiple heterologous PRRSV strains” was to evaluate the protective efficacy of porcine reproductive and respiratory syndrome virus 2 (PRRSV-2) modified live vaccines (MLV) in nursery pigs in a worst-case scenario where MLV does not match the genetic profile of the field isolate. This study deals with an interesting topic and fits with the journal's scope. This is a well-designed and well-written work, providing interesting results and novel results. Therefore, I suggest the acceptance under a major revision based on the following major and minor comments.
Comments for the authors
Abstract
- Please state clearly the aim of your study.
- Please summarize the information presented in the abstract. The text is over 200 words long.
- In the keywords place nursery pigs in the first place, it is the focus species of the study.
Response 1:
Thank you for your comments and thoughtful insights. We provide a point by point response below to your kind comments.
Response for abstract part
Thank you for your suggestions. We have made edits to the study aim in the short summary and placed Nursery Pigs as the first keyword.
Revised manuscript Line 16-34: The objective was to understand the impact when different Modified Live Virus (MLV) Porcine Reproductive and Respiratory Disease (PRRS) vaccines were given to piglets born to dams given a different MLV PRRS vaccine. 76,075, 2-week-old piglets vaccinated with US1-MLV were allocated into 4 groups. US1-MLV, US2-MLV, and US3-MLV groups were vaccinated with PRRSV-2 MLV including Ingelvac® PRRS MLV, HP-PRRSV-2 based MLV, and Prime Pac® PRRS, respectively. The NonVac group was left unvaccinated. Measurements were conducted on sera to look at PRRSV-specific antibody, serum neutralizing antibody (SN), and PRRSV RNA. Average daily gain (ADG) and survival rate were compared between treatment groups. Pigs vaccinated with US3-MLV displayed significantly lower mortality and higher ADG compared to all other groups despite not matching the field isolates genetically. We also observed a natural shift in PRRSV-2 isolates from Lineage 8.7 to Lineage 1 across the production flow. Our findings suggest genetic similarity between field viruses and vaccine strains should not be used as a sole predictor of field performance and that selection should be based on production performance and safety profile. The results also suggest that different MLVs can be used at different stages of production if needed without adverse impacts on production.
Materials and Methods
- Provide details of the vaccination program of the trial piglets.
- Provide more data on whether the piglets received any preventive treatment.
- Provide further details on the method of blood collection (needle size, venipuncture), the method of obtaining serum from the blood samples and who was responsible for the assessment of mortality, feed intake, survival rate and ADG.
- Has there been any sample size calculation?
Response for materials and methods part
We have revised the materials and methods section as you suggested by adding more details about vaccination program in trial piglets, medication for piglets, and the blood sampling procedure.
Firstly, Revised manuscript Line 184-194: The vaccination program includes quarterly whole-breeding-herd vaccination. Replacement gilts were vaccinated with 2 doses of the PRRSV-2 MLV (Ingelvac® PRRS MLV, Boehringer Ingelheim, Germany) at 18 and 22 weeks of age. Piglets were intramuscularly vaccinated once at 2 weeks of age. In addition, vaccination against M. hyopneumoniae and PCV2 was routinely performed in all piglets at 3 weeks of age. Pigs in all treatment groups had ad libitum access to a high-quality, three-phase nursery feeding program. Each phase of the nursery feed included a combination of tiamulin and amoxicillin at concentrations of 150 and 400 ppm, respectively, to control bacterial infections.
Secondly, Revised manuscript Line 217-218: A new sterile 1-inch, 18-gauge needle and syringe were used to collect a blood sample from each animal.
Thirdly, Revised manuscript Line 223-226: Survival rate and average daily gain (ADG) were collected from company data records across all animals and grow-outs enrolled in the study. The person responsible for assessing the mortality, survival rate, feed intake and ADG was the farm's supervising veterinarian.
For sample size calculation: A sample size calculation was not performed for this study due to logistical constraints related to the company we were working with. However, we aimed to collect at least 10 samples per experimental group, which aligns with a previous study (Park et al., 2020) that employed similar data collection methods.
Discussion:
- You could discuss the economic impact of your results on pig production.
- What are the negative points of the research, if any? Your recommendations for future studies, if any?
Response for discussion part
Firstly, thank you for your comment. We believe that we found a difference of 11% in survival rates and 54grams/day between US3-MLV, a strain that does not match the field strain and US2-MLV, a strain that matches the field strain. Based on this, we looked in the literature and estimated the value of this to be approximately USD $10.50 - $12.50.
Revised manuscript Line 493-499: Analyzing the results between US2-MLV and US3-MLV, we observed differences of 54grams/day and 11% survival rates. This represents a significant economic impact to the end user. Using other publications as a benchmark for instance, the study by Holtkamp et al21 found in US production conditions, a difference of 10% survival rates and similar reduction in average daily gain cost approximately $10.50 - $12.50 USD per pig marketed.
Secondly, thank you for your suggestion. We have written a separate segment on limitations.
Revised manuscript Line 608 – 618: Some limitations of this study are that we had to work with the logistics and delivery schedules of the company involved in the study. This meant that we were not able to homogenize pigs by antibody levels at the start of the study and this may have contributed to some confounding effects. As multiple grow-outs were involved, different factors may have influenced the survival rates of pigs, such as difference in staff training and motivation. We attempted to control for this by averaging results across different nursery grow-out sites. For example, Pigs in US1-MLV and US3-MLV were weaned at 26 days of age and separated into 56 and 32 different nursery grow-outs, respectively.
Comments on the Quality of English Language
- Moderate editing regarding English language is required for the improvement of the text.
Response for comments on the Quality of English Language
We have done some language editing. If still unsuitable, we will submit it for formal English language editing.

Round 2
Reviewer 1 Report
Comments and Suggestions for Authors
Regarding antibody response, there is something unclear. How to understand the description in P12 L422-423, “Pigs in the NonVac group remained serologically negative”. At DPV 0, Figure 4 showed similar antibody titter. In contrast, Figure 5A showed no antibody response at DPV 0. Furthermore, there was no any antibody response in the other DPV. In all, the authors must describe the results of antibody response clearly.
Author Response
Reviewer 1
Comments and Suggestions for Authors
Regarding antibody response, there is something unclear. How to understand the description in P12 L422-423, “Pigs in the NonVac group remained serologically negative”. At DPV 0, Figure 4 showed similar antibody titter. In contrast, Figure 5A showed no antibody response at DPV 0. Furthermore, there was no any antibody response in the other DPV. In all, the authors must describe the results of antibody response clearly.
Response to reviewer1:
Thank you for your comments and kind suggestions. We clarified the seronegative status of the NonVac group in Figure 5A. Additionally, we revised the results at point 3.5 in the results section, which are shown in the revised manuscript on Lines 469-486, highlighted in blue.
To better understanding, the NonVac group remained serologically negative throughout the experiment, as determined by the serum neutralization (SN) assay against the homologous virus, is because the homologous virus used in the experiment was a vaccine isolate. As a result, the unvaccinated group did not show any detectable SN titers.
Revised manuscript Line 469-486: Pigs in the NonVac group remained serologically negative throughout the experiment, as demonstrated by the serum neutralization (SN) assay against homologous viruses which refer to the vaccine isolates shown in Fig. 5A. In all vaccinated groups, the SN assay against homologous virus (Fig. 5A) showed a similar pattern of SN titers. At 0 DPV, SN titers in the US1-MLV were significantly (p<0.001) higher than those in the NonVac, US2-MLV and US3-MLV groups, however, the average SN levels in US2-MLV and US3-MLV groups were not significantly different. In all vaccinated groups, SN levels decreased slightly and reached their lowest level at 14 DPV. Subsequently, SN titers against the homologous virus gradually increased, peaking at 42 DPV, with the highest SN titers observed in the US1-MLV group. However, there was no significant difference in SN titers between the US1-MLV and US3-MLV groups post-vaccination. Although, the US2-MLV group was one of three vaccinated groups, the SN titer was still lower during the nursery period in the US2-MLV group compared to the US1-MLV and US3-MLV groups (Fig. 5A).

Reviewer 2 Report
Comments and Suggestions for Authors
Weakness:
The authors used pigs with significant differences in PRRSV initial specific and neutralizing antibodies to evaluate the protective effect of different vaccines by specific antibodies, neutralizing antibodies, ADG, and survival rate after immunization. The initial antibody levels of pigs are different, and the initial antibody levels, especially the neutralizing antibody levels, can affect the survival rate of piglets. Among the evaluation indexes, antibody level is one of the key indexes, and there are no correlation indexes of cellular immune response. The authors reply that they could not homogenize pigs by antibody levels due to the company's production requirements and would design trials to examine indexes of the cellular immune response in the future. However, those cannot be used to support the results of this study.
Specific points:
In the text, Fig.4 appears earlier than Fig.3, so it is recommended to swap it.
Author Response
|
Dear Reviewer,
Thank you very much for taking the time to review the revised manuscript. We have thoroughly considered your comments and would like to provide additional replies and information in response to your comments. Please find the detailed responses below and the corresponding revisions/corrections highlighted changes in the re-submitted files.
|
Reviewer 2
Comments and Suggestions for Authors
Weakness:
The authors used pigs with significant differences in PRRSV initial specific and neutralizing antibodies to evaluate the protective effect of different vaccines by specific antibodies, neutralizing antibodies, ADG, and survival rate after immunization. The initial antibody levels of pigs are different, and the initial antibody levels, especially the neutralizing antibody levels, can affect the survival rate of piglets. Among the evaluation indexes, antibody level is one of the key indexes, and there are no correlation indexes of cellular immune response. The authors reply that they could not homogenize pigs by antibody levels due to the company's production requirements and would design trials to examine indexes of the cellular immune response in the future. However, those cannot be used to support the results of this study.
Response to reviewer for weakness point:
We conducted an analysis of the cellular immune response as agreed upon. Specifically, we investigated the ability of the homologous virus (vaccine strain) to activate T lymphocytes by inducing the production of interferon-γ (IFN-γ) and interleukin-10 (IL-10). Revisions have been made to the manuscript, with updates to the materials and methods, figure, results, and discussion sections. These changes are highlighted in blue for ease of review.
For materials and methods section
Revised manuscript Line 205-209: Peripheral blood mononuclear cells (PBMC) were isolated and analyzed for interleukin 10 (IL-10) using commercial enzyme-linked immunosorbent assay (ELISA) at 14 DPV, and interferon γ (INF-γ) using enzyme-linked immunosorbent spot (ELISPOT) at 35 DPV, respectively.
Revised manuscript Line 299-326:
2.8. Analysis of lymphocytes cytokine production both INF-γ and IL-10
Heparinized blood was collected at 14 and 35 DPV. PBMC were isolated by layering the blood on a density gradient medium (LymphoSep™, MP Biomedicals, California, USA), followed by gradient centrifugation in accordance to the manufacturer's instructions. PBMC was then recalled with homologous virus (vaccine strain), and IFN-γ and IL-10 levels were measured as following.
2.8.1. Enzyme-linked immunospot (ELISPOT) assay
PBMC (2 × 105 cells/well) were incubated with either mock media, 0.01 MOI of homologous virus (vaccine strain) or PHA (10 μg/ml, Sigma-Aldrich, St. Louis, MO, USA) for 20 hr at 37ËšC in 5% CO2. IFN-γ production by the treated PBMC was assessed using an ELISPOT kit (R&D Systems, Minneapolis, MN, USA) according to the manufacturer's instructions. Spot-forming cells (SFCs) on the PVDF membrane were counted using an ELISPOT Reader (AID ELISPOT Reader, AID GmbH, Strassberg, Germany).
2.8.2. Quantification of IL-10 levels
IL-10 levels were measured from PBMC supernatants using porcine ELISA IL-10 kit (R&D Systems, Minneapolis, MN, USA) according to manufacturer’s protocol. Briefly, PBMC (2 × 106 cells/well) were culture with either 0.01 MOI of homologous virus (vaccine strain) or PHA (10 μg/ml, Sigma-Aldrich, St. Louis, MO, USA) for 20 hr at 37ËšC in 5% CO2. The culture supernatants were collected, and IL-10 levels were quantified by comparison to a porcine IL-10 standard. Optical density (OD) was measured at 450 nm using a microplate reader (Metertech, Taipei, Taiwan).
For figure section

Figure 6. Homologous virus (vaccine strain) induced IFN-γ- and IL-10 producing lymphocytes at 35 and 14 DPV, respectively. IFN-γ producing cells (A) and IL-10 levels in supernatant of stimulated PBMC cells (B) were measured. Data are demonstrated as mean ± standard error of the mean (SEM) from 5 pigs/group. Groups labeled with different letters indicate statistically differences between groups (p-value <0.01).
For results section
Revised manuscript Line 513-530:
3.6 Homologous virus as vaccine strain induced IFN-γ- and IL-10 producing lymphocytes
We investigated the ability of homologous virus (vaccine strain) to activate T lymphocytes by inducing production of IFN-γ and IL-10. The production of these cytokines were analyzed using commercial ELISPOT and ELISA kits. IFN-γ-producing lymphocytes were detected in all vaccination groups except for NonVac group (Fig. 6A). Amongst the vaccinated groups, the US3-MLV group exhibited the lowest levels of IFN-γ production, while the US2-MLV group showed the highest levels. However, the differences in IFN-γ production among the vaccinated groups were not statistically significant.
Similarly, supernatants from PBMC stimulated with the homologous virus (vaccine strain) showed the highest IL-10 levels in the US2-MLV group, significantly exceeding those in the other groups, followed by the US1-MLV group (Fig. 6B). The US3-MLV group had the significantly lowest IL-10 level. However, the IL-10 levels were not significantly different between the US1-MLV and US2-MLV groups.
For discussion section
Revised manuscript Line 598-605: The role of CMI in controlling PRRSV has been extensively studied in prior publications27-29. IFN-γ produced by activated T cells and macrophages, plays a crucial role in cellular immunity by facilitating inhibit viral replication, antigen presentation and phagocytosis30,31. In the present study, homologous virus stimulation significantly increased IFN-γ-producing lymphocytes in vaccinated groups, particulary in US2-MLV group compared to the NonVac group, aligning with findings from a previous study32.
Revised manuscript Line 614-631: Prior studies have demonstrated the immunosuppresive effect of IL-1033,34 and we assessed the levels induced by the different MLVs in this study. To provide a clearer assessment, IL-10 levels were measured at 14 DPV, based on a previous study that reported IL-10 levels to be highest at 14 DPV and gradually decreasing by 35 DPV32. Unsurprisingly, IL-10 levels were detected following homologous in vitro stimulation in all vaccinated groups, with the US2-MLV group showing significantly higher levels than the other vaccinated groups. Factors influencing IL-10 levels, such as the viral strain used in the experiment, could either upregulate or suppress IL-10 production35. Notably, elevated IL-10 levels can lead to reinfection or persistence of disease in the host, ultimately hindering PRRSV clearance36. This is consistent with the improved performance of US1-MLV and US3-MLV compared to US2-MLV. In our results, the US3-MLV group displayed the lowest levels of both IFN-γ and IL-10. These results suggest that US3-MLV does not stimulate a cytokine storm.Our findings demonstrate that using a homologous virus can effectively induce immune responses against PRRSV.
Specific points:
In the text, Fig.4 appears earlier than Fig.3, so it is recommended to swap it.
Response to reviewer for specific points:
Thank you for your suggestion. In the results section, we swapped the result shown in Fig.4 under the subtopic “Viremia (RT-PCR) in serum samples” with the result shown Fig.3 under the subtopic “Phylogenetic analysis of PRRSV-2 isolates”. All edited data were highlighted in blue.
Revised manuscript Line 369: 3.2. Phylogenetic analysis of PRRSV-2 isolates
Revised manuscript Line 412-418:
3.3. Viremia (RT-PCR) in serum samples
PRRSV-1 and PRRSV-2 were not detected in pigs from the US3-MLV group throughout the experiment. In contrast, PRRSV-1 and PRRSV-2 RNA were detected in pigs from the NonVac and US2-MLV groups at 14, 28, 42 and 56 DPV. The pigs in US1-MLV group, PRRSV-1 and PRRSV-2 RNA were detected early at 42 DPV and PRRSV-1 RNA was detected only at 56 DPV (Fig.4).